# Exceptional points of any order in a generalized Hatano-Nelson model

**Julius T. Gohsrich**[1,2,⋆], **Jacob Fauman**[1,2] **and Flore K. Kunst**[1,2,†]

**1** Max Planck Institute for the Science of Light, Staudtstraße 2, 91058 Erlangen, Germany
**2** Department of Physics, Friedrich-Alexander Universität Erlangen-Nürnberg,
Staudtstraße 7, 91058 Erlangen, Germany

⋆ julius.gohsrich@mpl.mpg.de ,    † flore.kunst@mpl.mpg.de

## Abstract

Exceptional points (EPs) are truly non-Hermitian (NH) degeneracies where matrices become defective. The order of such an EP is given by the number of coalescing eigenvectors. On the one hand, most work focuses on studying $N$th-order EPs in ($N \leq 4$)-dimensional NH Bloch Hamiltonians. On the other hand, some works have remarked on the existence of EPs of orders scaling with systems size in models exhibiting the NH skin effect. In this work, we introduce a new type of EP and provide a recipe on how to realize EPs of arbitrary order not scaling with system size. We introduce a generalized version of the paradigmatic Hatano-Nelson model with longer-range hoppings. The EPs existing in this system show remarkable physical features: Their associated eigenstates have support on a subset of sites and exhibit the NH skin effect, which can be tuned to localize on the opposite end of the chain compared to all remaining states. Furthermore, the EPs are robust against generic perturbations in the hopping strengths as well as against a specific form of on-site disorder.

## 1 Introduction

Non-Hermitian (NH) operators, while violating the axioms of quantum mechanics, have many applications in classical setups, such as electric circuits [1, 2] and optical metamaterials [3], while also being highly relevant for open quantum systems [4] and closed strongly correlated systems [5, 6]. In recent years, non-Hermiticity has been studied from the perspective of topology, revealing rich, novel phenomena and resulting in an exciting cross-disciplinary research field [7, 8].

While the conventional bulk-boundary correspondence (BBC) is generally broken in NH models and needs to be modified [9–11], an additional, truly NH BBC correspondence can be established, which directly relates the spectral topology under periodic boundary conditions (PBCs), captured by a spectral winding number [12], to the piling of bulk states on the boundaries under open boundary conditions (OBCs) [13–15], known as the NH skin effect (NHSE) [11]. This NHSE is always accompanied by the appearance of exceptional points (EPs) with an order scaling with system size [7, 16, 17]. EPs are truly NH degeneracies, at which the NH Hamiltonian is defective and whose order is set by the number of coalescing eigenvectors [18]. Indeed, it is straightforward to see how such an EP emerges in a system with skin states by considering the paradigmatic Hatano-Nelson (HN) model [19, 20]. In this nearest-neighbor hopping model with asymmetric hopping strengths all states pile up on the boundary as dictated by the dominant hopping parameter. In the extreme limit where one hopping is set to zero, all states coalesce onto one at the boundary thus forming an EP with an order scaling with the number of sites.

EPs are ubiquitous [21], and naturally appear in any NH system. In particular, it has been shown that symmetries can aid to find EPs of higher-order in lower-dimensional systems [22–

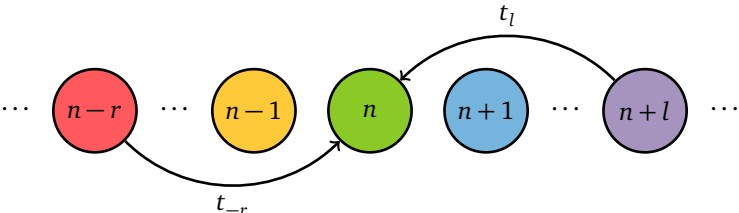

Figure 1: The generalized HN model with hoppings $t_l$ ($t_{-r}$) hopping $l$ ($r$) sites to the left (right). Each site contains its site index, and the model is $(l + r)$-partite. It reduces to the customary HN model for $l = r = 1$.

27]. In fact, it was recently pointed out that the much weaker condition of having a similarity has the same consequences [28]. All these studies mainly focus on the appearance of EPs of the order of the system size $N$. While a few remarks are made about the appearance of EP*m*s with $m < N$ in Refs. 25–27, and EP3s and EP4s are found in an SSH chain under OBC in Ref. 29, there is not yet a systematic study of how to generate EPs of any order $m$ in an $N$-dimensional system. In this work, we propose a method for finding such lower-order EPs by studying models akin to the HN model.

In particular, we study a family of generalized HN models, which only allow hoppings $l$ sites to the left and $r$ sites to the right as sketched in Fig. 1. The generalized HN model and similar models have mainly been studied in the thermodynamic limit in the mathematics [30, 31] and physics literature [32–35], especially in the context of the generalized Brillouin zone theory [11, 36, 37]. Here, we focus on features of these models for *finite* system sizes, and reveal a generic mechanism in which EP*m*s appear. Interestingly, while the appearance of such EPs depends on the system size $N$, its order does not scale with it. This behavior finds its root in a generalized chiral symmetry [38], which imposes a rotational symmetry in the spectrum shown in Fig. 2, pinning the EPs to the center of rotation.

We find that all eigenstates exhibit the NHSE, including the ones associated with the EPs, which are localized on a specific set of sites. Indeed, the system is $(l + r)$-partite so that we can subdivide the system into sublattices (SLs). Furthermore, we show that it is possible to localize the state associated with the EP and the remaining eigenstates on opposite ends of the chain. Lastly, we realize that the EPs are robust against generic perturbations in the hopping strengths [39, 40], and are thus protected by the spatial topology of the model. The EPs are also robust against a particular type of on-site disorder, which only exists on certain SLs. In the following, we discuss all of these features in detail.

## 2 The generalized Hatano-Nelson model

The family of generalized HN models we investigate, cf. Fig. 1, is described by

$$H_{lr} = \sum_{n=1}^{N} \left( t_l \, c_n^\dagger c_{n+l} + t_{-r} \, c_n^\dagger c_{n-r} \right), \tag{1}$$

where the chain has $N$ sites, $c_n$ ($c_n^\dagger$) annihilates (creates) an excitation on site $n$, the first (second) term describes the hopping of $l$ ($r$) sites to the left (right) with hopping strength $t_l$ ($t_{-r}$). Without loss of generality, we set $t_l, t_{-r} > 0$ and require $l \geq r \geq 1$ coprime, so that $l$ and $r$ have a greatest common divisor of one, i.e., $\gcd(l, r) = 1$, which we justify below. We note that whereas the customary HN model is NH iff $t_1 \neq t_{-1}$, the generalized HN model is always NH even when $t_l = t_{-r}$ as long as $l \neq r$.

Under PBCs, the Bloch Hamiltonian $H(k) = t_l e^{ilk} + t_{-r} e^{-irk}$ describes the generalized HN model. As a single-band model, it on the one hand directly corresponds to the energies, which form flowers as shown in gray in Fig. 2. On the other hand, this model cannot exhibit EPs under PBCs. In contrast, the spectrum under OBCs forms a star in the thermodynamic limit [30, 31, 34] as shown in light blue in Fig. 2, which exhibit EPs of any order as we show in the following.

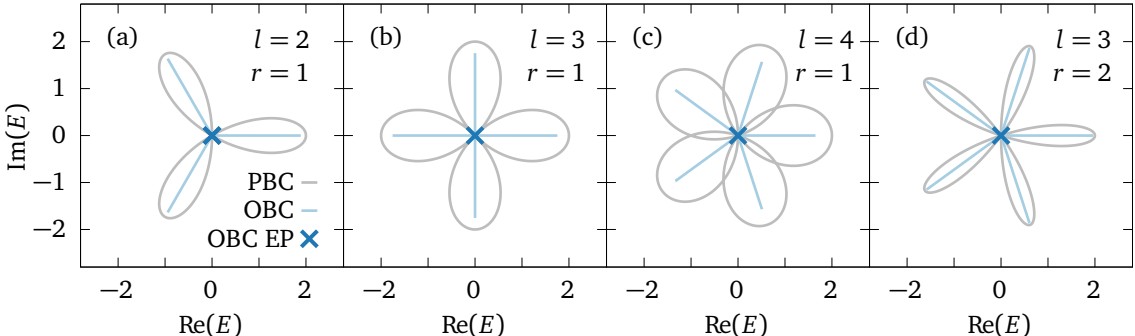

Figure 2: PBC (gray) and OBC (light blue) spectra of the generalized HN model in the thermodynamic limit with $t_l = t_{-r} = 1$ and $l$ and $r$ as indicated. Under OBCs with appropriate system size, we find an EP2, EP3, EP4, and EP4, in (a), (b), (c), and (d), respectively, marked with a dark blue cross, due to the $(l + r)$-fold spectral rotational symmetry.

## 3   Exceptional points of any order in the generalized HN Model

We consider a finite system under OBCs throughout this work unless stated otherwise. Let us focus on a paradigmatic example in the following before we return to the general case and finally give a recipe to find EPs of any order.

### 3.1   Example: $l = 2$ and $r = 1$

We consider $H_{21}$ shown in Fig. 3(a) with the characteristic polynomial given by

$$\chi(E) = (-E)^d \sum_{m=0}^{q} \binom{N - 2m}{m} \left(t_2 t_{-1}^2\right)^m (-E^3)^{q-m}, \tag{2}$$

where $N = 3q + d$ with $0 \le d < 3$, i.e., $q = \lfloor N/3 \rfloor$ is the quotient with $\lfloor . \rfloor$ the floor function and $d = N \bmod 3$ is the remainder of the Euclidean division, see Appendix A. The spectrum of $H_{21}$ is given by $\{E : \chi(E) = 0\}$, from which one can immediately read off spectral properties. While the factor $(-E)^d$ shows a $d$-fold degeneracy at zero energy, the $(-E)^3$ dependence dictates that the remaining eigenvalues come in triplets $\{E, E\omega_3, E\omega_3^2\}$ with $\omega_3 = e^{2\pi i/3}$. Thus, the complex spectrum of $H_{21}$ exhibits a 3-fold rotational symmetry as shown in Fig. 2(a). In Appendix B we show that the system has exactly $d$ zero-energy eigenvalues. In anticipation of the general case, we remark that the system is 3-partite. This implies we can define three SLs, $SL_1$, $SL_2$ and $SL_3$, shown in red, yellow and green in Fig. 3(a), where the site index $n$ satisfies $n \bmod 3 = 0$, 2 and 1, respectively. We emphasize that these three SLs should not be mistaken for degrees of freedom in the Bloch picture, instead they are signaling the presence of the spectral rotational symmetry.

Looking at the eigenspace structure of the $d$-fold degenerate zero-energy solutions, we uncover the following mechanism: For $d = 0$, there is no associated eigenspace, for $d = 1$ a single eigenvector exists, and for $d = 2$ one can readily construct a Jordan chain of length 2, i.e., there is an eigenvector $|v_1\rangle$ and a generalized eigenvector $|v_2\rangle$ satisfying $H_{21}|v_2\rangle = |v_1\rangle$, showing that the system exhibits an EP2. These vectors are given by

$$|v_1\rangle \propto \sum_{j=0}^{q} (-t)^{q-j} c_{3j+2}^\dagger |0\rangle, \qquad |v_2\rangle \propto (t_{-1})^{-1} \sum_{j=0}^{q} (q - j + 1)(-t)^{q-j} c_{3j+1}^\dagger |0\rangle, \tag{3}$$

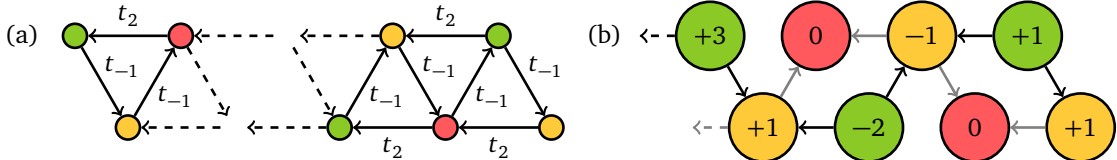

Figure 3: Generalized HN model for $l = 2$ and $r = 1$ under OBCs. (a) Alternative representation of Fig. 1 revealing the 3 SLs in red ($SL_1$), yellow ($SL_2$) and green ($SL_3$). (b) Visualization of the zero-energy eigenvector $|v_1\rangle$ (generalized eigenvector $|v_2\rangle$) for $t_2 = t_{-1} = 1$ on the yellow (green) SL with weights inside each node, forming the Jordan chain associated with the EP2 when the system size satisfies $N \equiv -1 \mod 3$. Acting with $H_{21}$ on $|v_1\rangle$ ($|v_2\rangle$) annihilates on $SL_1$ (creates $|v_1\rangle$ on $SL_2$) following the gray (black) arrows.

where $t = t_2/t_{-1}$, and are visualized in Fig. 3(b). From their form one can see that the zero-energy eigenvector $|v_1\rangle$ (generalized eigenvector $|v_2\rangle$) only has weight on the yellow (green) SL, and has no weight on the red SL. Furthermore, both $|v_1\rangle$ and $|v_2\rangle$ depend on the hopping ratio $t$, and are thus exponentially localized on the left (right) for $t_2 > t_{-1}$ ($< t_{-1}$) revealing a footprint reminiscent of the NHSE, which we further explore below.

In order to construct a zero-energy eigenvector in this or similar models, one can use a destructive interference argument discussed in Ref. 41, which in our case crucially depends on the right termination of the chain, which manifests itself in the periodicity of the generalized eigenspaces. Having such an eigenvector, all generalized eigenvectors can then be constructed in a straightforward fashion.

## 3.2 General case

In order to generalize to larger $l$ and $r$, we choose the matrix representation $\mathcal{H}_{lr}$ of Eq. (1) as

$$\mathcal{H}_{lr} = \begin{pmatrix} 0 & h_1 & 0 & \ldots & 0 \\ 0 & 0 & h_2 & \ddots & \vdots \\ \vdots & & \ddots & \ddots & 0 \\ 0 & & & \ddots & h_{l+r-1} \\ h_{l+r} & 0 & \ldots & \ldots & 0 \end{pmatrix}, \tag{4}$$

where the $h_j$ with $j = 1, 2, \ldots, l+r$ are rectangular matrices of size $d_j \times d_{j+1}$ with $d_{l+r+1} \equiv d_1$ describing the hopping from $SL_{j+1}$ to $SL_j$. We have chosen $l$ and $r$ coprime so that $\mathcal{H}_{lr}$ is $(l+r)$-partite, otherwise the system would split into $\gcd(l, r)$ decoupled subsystems, where each individual subsystem can again be treated using our formalism. For compactness, we drop the indices of $\mathcal{H}_{lr}$ when we consider arbitrary $l$ and $r$. We remark that a broad class of models with *Bloch Hamiltonians* of the form of Eq. (4) have been investigated in Ref. 38 in the context of *flat band physics*. While the mathematical structure is similar, the physical implications are vastly different. In the following, we review and adapt arguments of Ref. 38 to our problem, setting notation and providing additional insights.

The next step is to infer properties of $\mathcal{H}^{l+r}$ and map them back to $\mathcal{H}$. As $\mathcal{H}$ can be interpreted as a hopping model through its adjacency graph, raising $\mathcal{H}$ to the $n$th power corresponds to $n$ steps through the adjacency graph of $\mathcal{H}$. From Fig. 3(a) it is clear that $\mathcal{H}_{21}^3$ maps all states localized on $SL_j$ back to $SL_j$ for all $j$, so $\mathcal{H}_{21}^3$ is block diagonal, which is a general statement for all $l$ and $r$ taking $l + r$ steps. Thus, we write $\mathcal{H}^{l+r} = \text{diag}(\mathcal{H}_1, \mathcal{H}_2, \ldots, \mathcal{H}_{l+r})$ with $\mathcal{H}_j = h_j \cdot h_{j+1} \cdots h_{l+r} \cdot h_1 \cdots h_{j-1}$, where each block $\mathcal{H}_j$ with dimension $d_j \times d_j$ describes

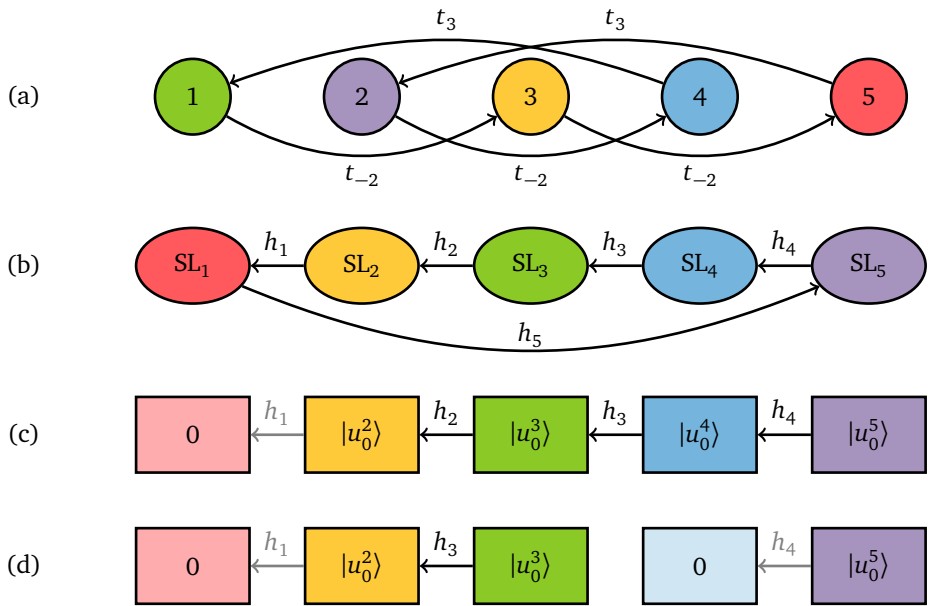

Figure 4: Determination of the EP structure of the generalized HN model with $l = 3$ and $r = 2$. (a) The hopping model in the site basis with corresponding site index inside each node and SL coloring corresponding to (b). Increasing $N$ one by one, the sizes of $SL_3$, $SL_5$, $SL_2$, $SL_4$, $SL_1$ are increased cyclically. Starting with $N \equiv 4 \bmod 5$ in (c), the generalized zero-energy eigenvectors form a Jordan chain of length 4 corresponding to an EP4. Decreasing the system size to $N \equiv 3 \bmod 5$ in (d) removes $|u_0^4\rangle$ on $SL_4$ and splits the Jordan chain into one of length 1, corresponding to a one-dimensional zero-energy eigenspace, and a Jordan chain of length 2 corresponding to an EP2. For $N \equiv 2 \bmod 5$, $N \equiv 1 \bmod 5$ and $N \equiv 0 \bmod 5$ one finds two, one and zero one-dimensional zero-energy eigenspaces of $\mathcal{H}$, respectively (not depicted).

a hopping model solely on $SL_j$, and without loss of generality we sort the SLs so that $d_1 \leq d_j$ for all $j$. The SL sizes $d_j$ are readily determined for all system sizes $N$. The small SLs such as $SL_1$ have size $d_1 = \lfloor N/(l+r) \rfloor$, whereas the large SLs have $d_j = \lceil N/(l+r) \rceil$, where $\lceil . \rceil$ is the ceiling function, such that $d_j = d_1 + 1$ if $N \bmod (l+r) \neq 0$.

In Ref. 38 it was shown that one can diagonalize all the blocks $\mathcal{H}_j$ in $\mathcal{H}^{l+r}$ as $\mathcal{H}_j |u_s^j\rangle = E_s |u_s^j\rangle$, $s = 1, 2, \ldots, d_1$, where the $E_s$ are the same for all $j$, and $E_s \neq 0$ for our model (cf. Appendix B). For all larger SLs with $d_j > d_1$, all remaining energies are zero, i.e., $\mathcal{H}_j |u_t^j\rangle = 0$, $t = d_1 + 1, \ldots, d_j$. In our case we have at most one zero-energy solution per SL and we relabel their corresponding eigenvectors as $|u_0^j\rangle$. Having the full spectrum of $\mathcal{H}^{l+r}$, the spectrum of $\mathcal{H}$ consists of $d = \sum_{j=2}^{l+r} (d_j - d_1)$ zero energies (cf. Appendix B), which is in our case the number of large SLs, i.e., $d = N \bmod (l+r)$, and the $(N-d)$ energies $\{\sqrt[n]{E_s}, \omega_n \sqrt[n]{E_s}, \ldots, \omega_n^{n-1} \sqrt[n]{E_s}\}$, where $s = 1, \ldots, d_1$, $\omega_n = e^{2\pi i/n}$ with $\omega_n^n = 1$ and $n = l + r$. This was shown in Ref. 38 by leveraging that Eq. (4) obeys a generalized chiral symmetry $\mathcal{C}_n : \Gamma_n \mathcal{H} \Gamma_n^{-1} = \omega_n^{-1} \mathcal{H}$, where the generalized chiral operator is $\Gamma_n = \mathrm{diag}(\mathbb{1}_{d_1}, \omega_n \mathbb{1}_{d_2}, \ldots, \omega_n^{n-1} \mathbb{1}_{d_n})$, with $\mathbb{1}_m$ the $m$-dimensional identity matrix, satisfying $\Gamma_n \Gamma_n^{-1} = \Gamma_n^n = \mathbb{1}_N$. In our previous example, we saw all these implications from the characteristic polynomial.

Besides these spectral considerations we analyze the eigenvectors of $\mathcal{H}$. First, it is instructive to define the padded eigenvectors $|\tilde{u}_s^j\rangle$ so that they are the eigenvectors of $\mathcal{H}^{l+r}$. The

eigenvectors $|w_s^p\rangle$ of $\mathcal{H}$ associated with $s \neq 0$, i.e., $E_s \neq 0$ are given by

$$|w_s^p\rangle \propto \sum_{v=0}^{l+r-1} \left(E_s e^{2\pi \mathrm{i}/p}\right)^{-\frac{v}{l+r}} \mathcal{H}^v |\tilde{u}_s^1\rangle, \tag{5}$$

where $p = 1, \ldots, l+r$ and we set $\mathcal{H}^0 = \mathbb{1}_N$. We note that a less compact form of this equation was also derived in the supplementary material of Ref. 42. Compared to the eigenvectors of $\mathcal{H}^{l+r}$, the $|w_s^p\rangle$ have weight on all SLs.

Coming to the zero-energy eigenvectors, we use that $\mathcal{H}^{l+r}|\tilde{u}_0^j\rangle = 0$, which shows that all $|\tilde{u}_0^j\rangle$ are on the one hand zero-energy eigenvectors of $\mathcal{H}^{l+r}$, while on the other hand they are generalized eigenvectors of $\mathcal{H}$ by definition. However, a priori it is not clear what the length $m \leq l+r$ of the associated Jordan chain defining an EP$m$ is. Let us assume a situation where $\mathrm{SL}_{j-1}$ and $\mathrm{SL}_{j+m+1}$ are of size $d_1$, and $\mathrm{SL}_j$, $\mathrm{SL}_{j+1}$, $\ldots$, $\mathrm{SL}_{j+m}$ are of size $d_1 + 1$. As $\mathcal{H}$ maps $|\tilde{u}_0^i\rangle$ to $|\tilde{u}_0^{i-1}\rangle$, we see that $\mathcal{H}|\tilde{u}_0^j\rangle = 0$ as there is no generalized zero-energy eigenvector on $\mathrm{SL}_{j-1}$ to map to. Thus, $|\tilde{u}_0^j\rangle$ is a proper eigenvector of $H$ and the end of a Jordan chain. To build up a Jordan chain, one needs to iteratively solve $\mathcal{H}|\tilde{u}_0^{j+i}\rangle = |\tilde{u}_0^{j+i-1}\rangle$ for $i = 1, \ldots, m$, which is equivalent to solving $h_{j+i-1}|u_0^{j+i}\rangle = |u_0^{j+i-1}\rangle$. In Appendix C we show that in the described situation, $h_{j+i-1}$, $i = 1, \ldots, m$, is invertible, thus it is always possible to solve these equations. Finally, $h_{j+m}|u_0^{j+m+1}\rangle = |u_0^{j+m}\rangle$ is not solvable because there exists no generalized zero-energy eigenvector on $\mathrm{SL}_{j+m+1}$, such that $|\tilde{u}_0^{j+m}\rangle$ marks the start of a Jordan chain. We thus identified the Jordan chain $|\tilde{u}_0^{j+m}\rangle, \ldots, |\tilde{u}_0^{j+1}\rangle, |\tilde{u}_0^j\rangle$ of length $m$ and thus an EP$m$. Therefore, determining the lengths of all Jordan chains, i.e., the orders of all EPs, reduces to counting the number of consecutive large SLs. We remark that this procedure only depends on the existence of the zero-energy eigenvectors of the $\mathcal{H}_j$, and thus on $N \bmod (l+r)$ and not directly on the system size $N$. Fig. 4 shows how to determine the Jordan chains for $l = 3$ and $r = 2$ graphically. For completeness, we define $|w_0^p\rangle = |\tilde{u}_0^p\rangle$ so that the $|w_s^p\rangle$ are all (generalized) eigenvectors of $\mathcal{H}$.

## 3.3 General recipe towards finding EPs

Equipped with this algorithm we show how to engineer arbitrary low-order EP$m$s in the generalized HN model of size $N$. First, for $N \equiv -1 \bmod (l+r)$ we have the $l+r-1$ generalized eigenvectors $|\tilde{u}_0^j\rangle$ with $j = 2, \ldots l+r$ forming a Jordan chain of length $l+r-1$ corresponding to an EP$(l+r-1)$ as shown in the example in Fig. 4(c). Conversely, one can design a generalized HN model exhibiting an EP$m$ by choosing $l+r = m+1$, where $l > r \geq 1$ coprime, and system size $N$ so that $N \equiv -1 \bmod (l+r)$.

Secondly, we can simplify this further by choosing $r = 1$. From the previous paragraph we know that the system can host up to EP$l$s for $N \equiv -1 \bmod (l+1)$. However, decreasing the system size one by one removes subsequently $|\tilde{u}_0^{l+1}\rangle$ down to $|\tilde{u}_0^2\rangle$, shortening the Jordan chain one by one and thus reducing the order of the EP one by one as shown in Fig. 5. Conversely, one can engineer an EP$m$ by choosing any $l \geq m$ and $r = 1$ and choose a system size satisfying $N \equiv m \bmod (l+1)$.

The generalized HN is not restricted to featuring a single EP as one can have more elaborate zero-energy eigenspaces as already seen in the example $l = 3$ and $r = 2$ in Fig. 4(d). One can also get multiple EPs, e.g., when considering $l = 5$ and $r = 2$, one can find two EP2s for $N \equiv 4 \bmod 7$, and an EP2 and EP3 for $N \equiv 5 \bmod 7$.

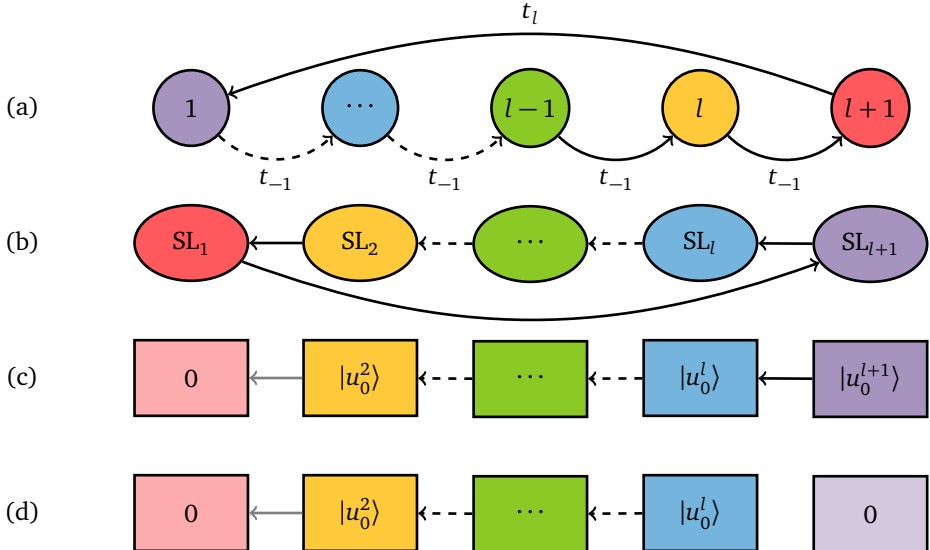

Figure 5: Determination of the EP structure of the generalized HN model for arbitrary $l$ and $r = 1$ similar to Fig. 4. (a) Increasing the chain length increases the sizes of SLs cyclically in (b). Starting with $N \equiv -1 \bmod (l + 1)$ corresponds to having an EP$l$ in (c). Successively decreasing the size of the chain always removes the generalized eigenvector with the highest SL index as shown in (d).

## 4 Properties of the EPs in the generalized HN model

Having established that the generalized HN model can host EPs of arbitrary order for an appropriate choice of $l$ and $r$, we want to determine further properties of their associated eigenvectors.

### 4.1 EPs exhibiting the NHSE

As extensively discussed in the literature, the NHSE is directly related to the spectral topology of NH tight-binding models, where the topological index is the spectral winding number [12]

$$w(E_{\mathrm{R}}) = \frac{1}{2\pi\mathrm{i}} \int_{-\pi}^{\pi} \mathrm{d}k \frac{\mathrm{d}}{\mathrm{d}k} \ln[H(k) - E_{\mathrm{R}}], \qquad (6)$$

where $E_{\mathrm{R}}$ is a reference energy and $H(k)$ is the Bloch Hamiltonian. The sign of the winding number predicts that the eigenstate associated with $E_{\mathrm{R}}$ is exponentially localized to the left (right) of the system when $\mathrm{sgn}\, w > 0$ ($\mathrm{sgn}\, w < 0$), where we note that an eigenstate is delocalized if its associated winding number is ill-defined, corresponding to a Bloch point [43].

We find that the correspondence is valid for all eigenvectors of the system, including the eigenvectors associated with the EPs. In the example $l = 2$ and $r = 1$, shown in Fig. 6, one can on the one hand determine the winding number at zero energy as $w(0) = 2$ ($-1$) if $t_2 > t_{-1}$ ($< t_{-1}$). On the other hand, the explicit form of the eigenvector $|v_1\rangle$ in Eq. (3) only depends on powers of $t_2/t_{-1}$. Thus, the sign of $w(0)$ correctly predicts the occurrence and exponential localization of the NHSE associated with that state.

In that example, it is also interesting to notice that one can always tune $t_2$ and $t_{-1}$ so that for a fixed $s \in [0, d_1]$, all eigenvectors associated with $|\sqrt[3]{E_s}| < |E_{\mathrm{B}}|$ are localized on one end of the chain, while the remaining eigenvectors with $|\sqrt[3]{E_s}| > |E_{\mathrm{B}}|$ are localized on the opposite end, where $E_{\mathrm{B}}$ is the energy associated with the aforementioned Bloch point [43], i.e., a self-intersection of the PBC spectrum, separating regions of positive and negative winding

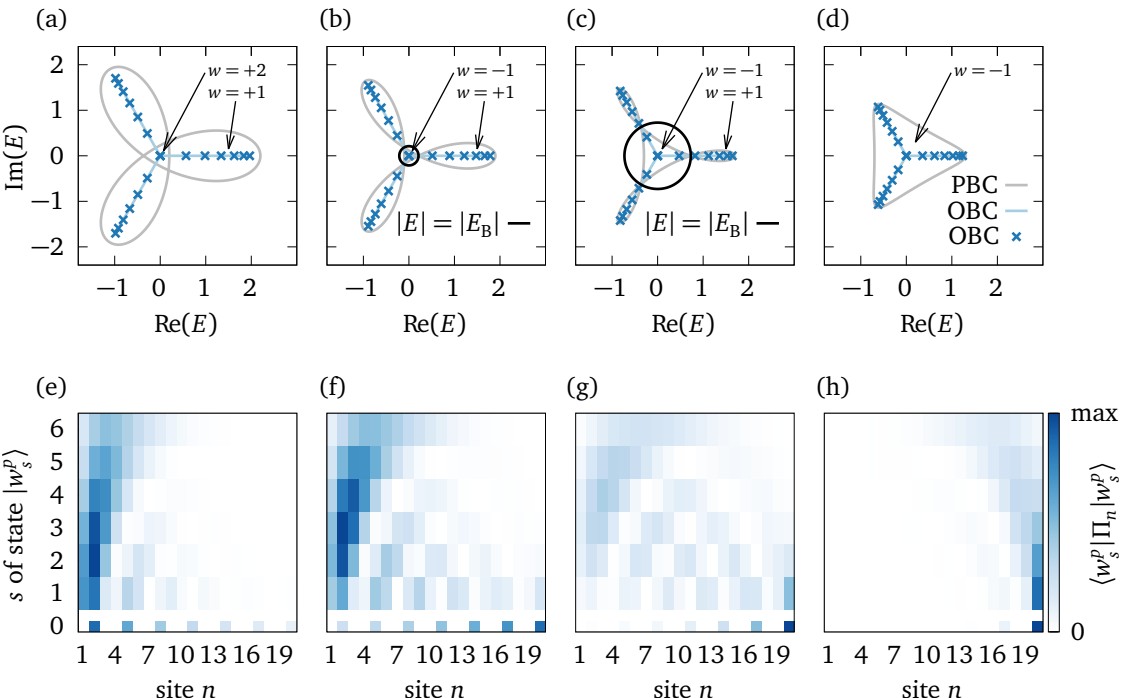

Figure 6: (a-d) PBC (light gray) and OBC (light blue) spectra and eigenvectors of the generalized HN model with $l = 2$, $r = 1$ and $t_{-1} = 1$ for different hopping strengths $t_2$. OBC spectra for finite size (dark blue crosses) and associated eigenvectors (blue scale) always correspond to a system of size $N = 20$ so that the system exhibits an EP2. All eigenvectors in (e-h) are normalized so that $\langle w_s^p | w_s^p \rangle = 1$ where $\langle w_s^p | = (|w_s^p\rangle)^\dagger$. (a) Spectra for $t_2 = 12/10 > t_{-1}$ showing only regions of positive winding number, predicting that all eigenvectors are localized on the left in (e). (b,c) Spectra for $t_2 = 9/10$ and $t_2 = 7/10$, respectively, showing regions with winding numbers $w = \pm 1$. These regions are separated by Bloch points, which are self-intersections in the PBC spectrum at $\{|E_B|, |E_B|\omega_3, |E_B|\omega_3^2\}$ with $\omega_3 = e^{2\pi i/3}$. We draw a black circle with radius $|E_B|$ to distinguish energies outside the circle having winding number $+1$, which associated eigenvectors localize on the left, from energies inside with $w = -1$, which associated eigenvectors are localized on the right. Thus, in (f) there is only the eigenvector associated with the EP, and in (g) additionally the eigenvectors labeled by $s = 1$, localized on the right, while all other states are localized on the left. (d) Spectrum for $t_2 = 3/10 < t_{-1}/2$ showing only a region with negative winding number, corresponding to all eigenvectors localized on the right in (h). (e-h) Eigenvectors associated with (a-d). To depict the localization of the eigenvectors $|w_s^p\rangle$, we plot $\langle w_s^p | \Pi_n | w_s^p \rangle$, where $\Pi_n = c_n^\dagger |0\rangle\langle 0| c_n$ is the projector onto each site in the chain. As such, for the eigenvectors associated with a non-zero energy, the phases depending on $p$ in Eq. (5) drop out and those eigenvectors are displayed in groups of three. The maximum values max of the color map are max $\approx 0.38, 0.28, 0.51, 0.91$, respectively.

numbers. The Bloch point, whose associated energy is purely real, can be determined by requiring $\mathrm{Im}(E_B) = \mathrm{Im}[H(k_B)] = 0$, which is solved by $k_B = 2\arctan[\sqrt{(2t_2 - t_{-1})/(2t_2 + t_{-1})}]$ if $t_{-1} < 2t_2$. In the thermodynamic limit, the maximum eigenvalue of $\mathcal{H}_{21}$ is given as $E_{\max} = 3t_2^{1/3}(t_{-1}/2)^{2/3}$ [30, 34], so the largest eigenvalue for finite system sizes is lower than that. As one has $0 \le E_B = t_2[(t_{-1}/t_2)^2 - 1] \le E_{\max}$, we can always tune $t_2$ and $t_{-1}$ appropriately. We can especially separate the eigenvector associated with the EP from the rest of the eigenvectors as shown in Figs. 6(b,f).

## 4.2 Robustness of the EPs and the NHSE

Now, let us review the robustness of the EPs against perturbations. Even though EPs are fine-tuned structures which usually break when perturbing [7, 21], we find two types of perturbations, which leave the EPs intact, namely, generic perturbations to the hopping strengths and arbitrary on-site disorder on specific SLs. We discuss these two types of perturbations, also with respect to the NHSE, in the following.

### 4.2.1 Disorder in hopping strengths

Regarding the disorder in the hopping strengths, we remind ourselves that the generalized chiral symmetry only depends on the form Eq. (4), thus perturbing the hoppings $t_a \to t_{a,n}$, $a = l, -r$, does not break this symmetry. As such, the occurrence and order of the EPs only depends on the sizes of the SLs and is thus protected by the topology of the adjacency graph. If this topology is unaltered, i.e., $t_{a,n} \ne 0$ for all $n$, the EPs stay unaltered. For a change in the graph topology, i.e., setting some $t_{a,n} = 0$, the matter is more subtle. For example, splitting the system in smaller ones can leave the EP unchanged, e.g., removing all hoppings from and to the first red and yellow site in Fig. 3(a) splits the system of size $N$ with $N \bmod 3 = 2$ into subsystems of size $N_1 = 3$ and $N_2 = N - 3$, where the former subsystem does not introduce new zero-energy solutions and the latter subsystem still exhibits an EP2. Another example would be to remove all the hoppings from a red site to green one via $t_2$ in Fig. 3(a). Therefore, the generalized chiral symmetry shows a slightly different characteristic compared to the more conventional NH symmetries, where symmetry-preserving perturbations keep the EPs in general intact [22–27].

In any case, the occurrence of the NHSE crucially depends on the specific values of the $t_{a,n}$. We find that introducing a random perturbation in the hoppings as $t_{a,n} = t_a(1 + \Delta_{a,n})$ with $\Delta_{a,n}$ uniform in $[-\Delta_a, +\Delta_a]$ does not destroy the NHSE for slight hopping disorders $\Delta_a$, an insight carrying over from the customary HN model [44, 45]. A spectrum and its associated eigenvectors for a realization of such a random perturbation is shown in Figs. 7(a,b).

### 4.2.2 On-site disorder

Let us now consider the second type of perturbation, on-site disorder. While the NHSE has been shown to be robust against on-site disorder up to a certain threshold as result of the spectral topology in case of the customary HN model [12, 44, 45], EPs are not known to be stable against such perturbations. However, for the generalized HN model we showed that all generalized eigenvectors forming a Jordan chain associated with a specific EP have weight only on specific SLs (in the example $l = 2$ and $r = 1$ on $SL_2$ and $SL_3$), but not on others ($SL_1$). Thus, any perturbation on the latter SLs will not affect the occurrence or order of that EP, even though it breaks the generalized chiral symmetry of $\mathcal{H}$. We show an example for $l = 2$ and $r = 1$ with random on-site disorder on $SL_1$ modeled by $H_{\mathrm{pert}} = \sum_n t_{0,n} \delta_{n \bmod 3, 0} c_n^\dagger c_n$ where $t_{0,n}$ uniform in $[-W, +W]$ depicted in Figs. 7(a,c). The robustness of the zero-energy eigenvector parallels the robustness of a zero-energy edge state in the (NH) SSH model [10, 46, 47], which

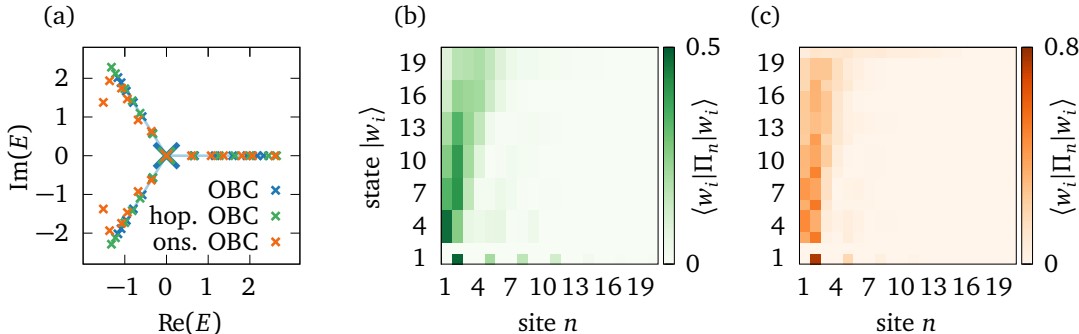

Figure 7: OBC spectra and eigenvectors of the generalized HN model with $l = 2$, $r = 1$, $t_2 = 2$ and $t_{-1} = 1$ and different perturbations. (a) Spectrum without perturbation (blue), with random perturbation in the hopping strengths characterized by $\Delta_l = \Delta_{-r} = 1/2$ (green), and random on-site disorder on $SL_1$ characterized by $W = 2$ (orange). In all cases, the EP2 at $E = 0$ is robust against the perturbation. (b,c) Eigenvectors, presented as in Fig. 6, associated with the random perturbations in the hopping and random on-site disorder on $SL_1$, respectively, corresponding to the spectrum in (a). One can clearly see that the perturbations do not alter the NHSE.

also has support on a single SL due to the conventional (NH) chiral symmetry, and is thus robust against on-site disorder on the other SL. For EPs, however, one also have to consider the support of the generalized eigenvectors to keep the Jordan chains intact.

Not only is the EP robust against this form of perturbation, but one can also use on-site disorder as a mechanism to reduce the order of an EP by altering its Jordan chain. For example, for $l = 3$, $r = 1$ and $N \equiv -1 \mod 4$ one has an EP3 with associated Jordan chain $H_{31}|\tilde{u}_0^4\rangle = |\tilde{u}_0^3\rangle$, $H_{31}|\tilde{u}_0^3\rangle = |\tilde{u}_0^2\rangle$ and $H_{31}|\tilde{u}_0^2\rangle = 0$. We introduce on-site disorder on $SL_4$ on which initially only $|\tilde{u}_0^4\rangle$ has weight, as $H_{\mathrm{pert}} = \sum_n t_{0,n} \delta_{n \bmod 4,1} c_n^\dagger c_n$, with $\delta$ the Kronecker delta. We find that $|\tilde{u}_0^4\rangle$ is no longer a generalized eigenvector as $(H_{31} + H_{\mathrm{pert}})|\tilde{u}_0^4\rangle = |\tilde{u}_0^3\rangle + H_{\mathrm{pert}}|\tilde{u}_0^4\rangle \neq |\tilde{u}_0^3\rangle$. Introducing such an on-site term shifts one eigenvalue away from zero while keeping the remainder of the Jordan chain, thus reducing the EP3 to an EP2. Introducing on-site disorder on SLs associated with generalized eigenvectors within a Jordan chain, e.g., on $SL_3$ where $|\tilde{u}_0^3\rangle$ has weight is more subtle: One might falsely guess that this splits the EP3 into two one-dimensional zero-energy eigenspaces plus another non-zero eigenspace. However, in that example it is possible to construct a new generalized eigenvector $|v\rangle$ with weight on $SL_3$ and $SL_4$, which satisfying $(H_{31} + H_{\mathrm{pert}})|v\rangle = |\tilde{u}_0^2\rangle$, showing that the perturbed system still exhibits an EP2, cf. Appendix D.

## 5 Conclusion

In this work, we introduced the generalized HN model, where setting the hopping ranges $l$ and $r$ to the left and right, respectively, allows generating EPs of arbitrary order under OBCs for appropriate system sizes. In contrast to previously studied unidirectional models, the EPs we find do not scale with system size, while their existence does crucially depend on the system size. To the best of our knowledge, these type of system-size dependent EPs with system-size independent orders have not been systematically studied so far.

We find that the EPs in our system show remarkable features. Firstly, the eigenstates corresponding to the EPs are localized on a subset of sites we identified as SLs, independent of their hopping strengths. Tuning these hopping strengths, we are able to manipulate the NHSE so that the eigenstates associated with the EPs localize on a different end as compared to the

remaining eigenstates of the system. Furthermore, as a result of the generalized chiral symmetry, the EPs are robust against generic perturbations in the hopping strength thus signaling that their occurrence finds its root in the spatial topology of the model. When we break the generalized chiral symmetry by introducing on-site disorder on specific SLs the EPs are either left unchanged or demoted in their order. We find that the NHSE does not vanish for any of the aforementioned perturbations for small perturbation strengths.

Furthermore, one can think of our generalized HN model as a sweet-spot in a much larger space of systems including multiple hoppings in all directions. The low-order EPs are then reached by appropriately tuning the system parameters, as is the standard procedure for going to EPs in any model. Besides the low-order EPs discussed in the main text, the generalized HN model exhibits another type of EP, which occurs when relaxing the constraint $t_l, t_{-r} > 0$ to also allow vanishing hopping strengths. Setting $t_{-r} = 0$ ($t_l = 0$) the generalized HN model decouples into $l$ ($r$) unidirectional chains corresponding to EPs scaling with system size, which can be seen in Fig. 3(a) for $t_{-1} = 0$ ($t_2 = 0$).

We emphasize that the methods developed in this work are applicable to any other model under OBCs, which can be brought into the form of Eq. (4), to find lower-order EPs in an intuitive way by determining the SL imbalances and Jordan chains. In this context, it is especially relevant to mention that the robustness against generic perturbations in the hopping strengths as well as the robustness against on-site disorder on specific SLs stays a feature in such models.

In this work we inferred the spectrum and eigenvectors of $\mathcal{H}$ from $\mathcal{H}^{l+r=n}$, which is the parent Hamiltonian in the context of $n$th-root topological phases [38,48–52]. Deeper connections, such as how the spectral topology of both models is connected, fall outside the scope of this work, and remain an open question. Another fascinating direction is an analysis of our model in the context of topological graph theory. We find that the generalized HN model under OBCs (PBCs) can always be embedded onto a cylinder (torus). As such, our work is connected to so-called helical lattices [53–59].

Our generalized HN model can readily be implemented in experiment. There are several platforms, which allow for the implementation of unidirectional couplings, such as photonic ring systems [42], topoelectric circuits [1,2], single-photon interferometry experiments simulating non-unitary quantum walks [60], and fiber loops modeling synthetic frequency dimensions [61,62]. The realization of our model in the lab would allow for a rigorous study of the properties of EPs unaffected by perturbations.

*Note added.* Shortly after the appearance of our work, a related paper appeared [63]. While Ref. 63 develops a topological categorization of systems with generalized chiral symmetry assuming SLs of the same size, our work focuses on a specific model with SLs of different sizes, and studies the resulting EPs and their physical properties.

## Acknowledgements

J.F. thanks Quentin Levoy for thoughtful discussions.

**Funding information**    J.T.G. and F.K.K. acknowledge funding from the Max Planck Society Lise Meitner Excellence Program 2.0. J.T.G. and F.K.K. also acknowledge funding from the European Union via the ERC Starting Grant "NTopQuant" (101116680). Views and opinions expressed are however those of the authors only and do not necessarily reflect those of the European Union or the European Research Council (ERC). Neither the European Union nor the granting authority can be held responsible for them. J.F. thanks the Max Planck School of Photonics for their generous support during his Master studies.

# A The characteristic polynomial for $l = 2$ and $r = 1$

We prove the form of the characteristic polynomial of $H_{21}$, Eq. (2) of the main text in three steps: First we write down a linear recurrence relation, secondly, we solve it in terms of generating functions, and finally, we rewrite this solution in the form presented in the main text.

To write down a linear recurrence relation for the characteristic polynomial, we choose the $N$-dimensional matrix representation $\mathcal{H}_{N,21}^{s}$, where the label s signifies that we write it in the site basis,

$$\mathcal{H}_{N,21}^{s} = \begin{pmatrix} 0 & 0 & t_2 & & & & & \\ t_{-1} & 0 & 0 & t_2 & & & & \\ & t_{-1} & 0 & 0 & t_2 & & & \\ & & \ddots & \ddots & \ddots & \ddots & & \\ & & & t_{-1} & 0 & 0 & t_2 \\ & & & & t_{-1} & 0 & 0 \\ & & & & & t_{-1} & 0 \end{pmatrix}. \tag{A.1}$$

Then

$$\chi_N(E) = \det\left(\mathcal{H}_{N,21}^{s} - EI_N\right)$$

$$= (-E)\det\left(\mathcal{H}_{N-1,21}^{s} - EI_{N-1}\right) + t_2 \det\begin{pmatrix} A & B \\ 0 & \mathcal{H}_{N-3,21}^{s} - EI_{N-3} \end{pmatrix}. \tag{A.2}$$

To find the equality, we use the Laplace expansion along the first row for the second equality, and

$$A = \begin{pmatrix} t_{-1} & -E \\ 0 & t_{-1} \end{pmatrix}, \qquad B = \begin{pmatrix} t_2 & 0 & 0 & \cdots & 0 \\ 0 & t_2 & 0 & \cdots & 0 \end{pmatrix}. \tag{A.3}$$

Using a determinant identity for block matrices

$$\det\begin{pmatrix} A & B \\ 0 & D \end{pmatrix} = \det(A)\det(D) = \det\begin{pmatrix} A & 0 \\ C & D \end{pmatrix}, \tag{A.4}$$

where $A$, $B$, $C$ and $D$ are rectangular blocks, we can immediately determine the second determinant to find

$$\chi_N(E) = (-E)\chi_{N-1}(E) + t_2 t_{-1}^2 \chi_{N-3}(E). \tag{A.5}$$

As the recurrence relation has an $N-3$ dependence we need to determine three base cases. They are

$$\chi_1(E) = (-E)^1, \qquad \chi_2(E) = (-E)^2, \qquad \chi_3(E) = (-E)^3 + t_2 t_{-1}^2. \tag{A.6}$$

Even though it seems nonsensical to define the characteristic polynomial for $N = 0$, it will be useful to define $\chi_0(E) = (-E)^0 = 1$, which is consistent with the recurrence relation and $\chi_3(E)$ from the previous equation, and use $\chi_0(E)$, $\chi_1(E)$ and $\chi_2(E)$ as base cases.

The next step is to find a generating function for $\chi_N(E)$ satisfying

$$S(x, E) = \sum_{N=0}^{\infty} \chi_N(E)x^N, \tag{A.7}$$

so that

$$\chi_N(E) = \frac{1}{N!} \left.\frac{d^N S(x, E)}{dx^N}\right|_{x=0}. \tag{A.8}$$

Multiplying the recurrence relation by $x^N$ and summing over $N$ we find an equation for $S(x, E)$,

$$\sum_{N=3}^{\infty} \chi_N x^N = \sum_{N=3}^{\infty} (-E\chi_{N-1} + T\chi_{N-3}) x^N, \tag{A.9}$$

where $T = t_2 t_{-1}^2$, and we start the sum at $N = 3$ for reasons that become apparent below, and we drop the $E$ dependence of $\chi$ for readability. After some index shifts, we have

$$\sum_{N=3}^{\infty} \chi_N x^N = -Ex \sum_{N=2}^{\infty} \chi_N x^N + Tx^3 \sum_{N=0}^{\infty} \chi_N x^N. \tag{A.10}$$

To get back $S(x, E)$, we subtract and add the appropriate terms as

$$\sum_{N=0}^{\infty} \chi_N x^N = \sum_{N=0}^{m-1} \chi_N x^N + \sum_{N=m}^{\infty} \chi_N x^N \tag{A.11}$$

to find

$$S(x, E) - \sum_{N=0}^{2} \chi_N x^N = -Ex \left[ S(x, E) - \sum_{N=0}^{1} \chi_N x^N \right] + Tx^3 S(x, E). \tag{A.12}$$

Using the base cases $\chi_n = (-E)^n$ for $n = 0, 1, 2$ and rearranging we find the generating function

$$S(x, E) = \frac{1}{1 + Ex - Tx^3}. \tag{A.13}$$

Finally, we want to prove that the generating function $S(x, E)$ generates Eq. (2), which we repeat in a slightly different form here

$$\chi_N(E) = \sum_{m=0}^{\lfloor N/3 \rfloor} \binom{N - 2m}{m} \left( t_2 t_{-1}^2 \right)^m (-E)^{N-3m}. \tag{A.14}$$

Setting $N = 3q + d$ via Euclidean division proofs Eq. (2). We can expand the generating function using the geometric series as

$$S(x, E) = \frac{1}{1 - (-Ex + Tx^3)} = \sum_{n=0}^{\infty} \left( -Ex + Tx^3 \right)^n = \sum_{N=0}^{\infty} \chi_N(E) x^N. \tag{A.15}$$

To determine the characteristic polynomial we need to match terms. We start by considering the coefficient of $E^k$ multiplying $x^N$, i.e., the coefficient of $E^k$ in $\chi_N(E)$. First off, notice that unless $k \equiv N \bmod 3$, the coefficient will be 0. This is because all the terms will be products of $-Ex$ and $Tx^3$, and multiplying an expression by $-Ex$ increases the exponent of both $E$ and $x$ by 1, while multiplying by $Tx^3$ raises the exponent of $x$ by 3. Next, note that the coefficient of $E^N$ multiplying $x^N$ is simply $(-1)^N$, since the only product which achieves an $E^N x^N$ term is $(-Ex)^N$. The coefficient of $E^{N-3}$ is $T(-1)^{N-1}(N-2)$. This is because $E^{N-3}x^N$ is achieved by multiplying $(N-3)$ terms of $-Ex$ with 1 term of $Tx^3$. There are $(N-2)$ terms in total, so there are $\binom{N-2}{N-3} = N - 2$ ways to order them. The term with exponent $E^{N-3m}x^N$ is achieved by multiplying $(N - 3m)$ terms of $-Ex$ with $m$ terms of $Tx^3$. There are $(N - 2m)$ terms in total, and therefore $\binom{N-2m}{m}$ different ways to form a product with exponent $E^{N-3m}x^N$. The coefficient is therefore $T^m(-1)^{N-3m}\binom{N-2m}{m}$, concluding the proof.

A similar analysis could find expressions for any rational function in terms of binomial coefficients.

# B $\mathcal{H}_1$ does not have zero-energy eigenvalues

As discussed in Ref. 38 for general $n$-partite models with generalized chiral symmetry, the number of zero-energy solutions of $\mathcal{H}$ is given by $\sum_{j=2}^{n}(d_j - d_1) + n \#_0^{\mathcal{H}_1}$, where $\#_0^{\mathcal{H}_1}$ is the number of zero-energy solutions of the smallest block $\mathcal{H}_1$ of $\mathcal{H}^n$.

For the generalized HN model in our work, we never find a zero-energy solution in $\mathcal{H}_1$, i.e., $\#_0^{\mathcal{H}_1} = 0$. The intuition is that the spectrum of $\mathcal{H} \equiv \mathcal{H}_{lr}$ under OBCs in the thermodynamic limit forms a star [30, 31, 34], where a real arc of the star is the interval $[0, E_{\max}]$, $E_{\max} \in \mathbb{R}$, and all other arcs can be constructed by rotations in complex plane. Then, the spectrum for each SL, i.e., the spectra of $\mathcal{H}_1, \mathcal{H}_1, \ldots, \mathcal{H}_{l+r}$, are given by the interval $[0, E_{\max}^{l+r}]$. Now, for finite system sizes it is generally the case that the end points of the spectra in the thermodynamic limit are not in the spectrum for finite system sizes. Thus, any finite $H_j$ does not have a zero-energy eigenvalue, and any zero-energy eigenvalue of $\mathcal{H}_{lr}$ is a finite-size effect due to the interplay between the different SLs.

To underline the intuition of that end points of the spectra in the thermodynamic limit are not part of the spectrum for finite size, consider for example the customary HN model ($l = r = 1$), where the OBC spectrum is a real line with the interval $[-2\sqrt{t_1 t_{-1}}, 2\sqrt{t_1 t_{-1}}]$. For finite system size $N$, the spectrum is given by $2\sqrt{t_1 t_{-1}} \cos[m\pi/(N+1)]$, $m = 1, 2 \ldots, N$, such that for finite $N$ the end points of the OBC spectrum are never reached.

Let us now present an explicit proof for our main example. We show in the following that $\mathcal{H}_1$ for $l = 2$ and $r = 1$ has no zero-energy solutions. To do so, we derive its determinant, which is non-zero. $\mathcal{H}_1$ is given by

$$
\mathcal{H}_{1,N} = \begin{pmatrix}
3\,t_{-1}^2 t_2 & 3\,t_{-1}t_2^2 & t_2^3 & & & \\
t_{-1}^3 & 3\,t_{-1}^2 t_2 & 3\,t_{-1}t_2^2 & t_2^3 & & \\
& \ddots & \ddots & \ddots & \ddots & \\
& & t_{-1}^3 & 3\,t_{-1}^2 t_2 & 3\,t_{-1}t_2^2 & t_2^3 \\
& & & t_{-1}^3 & 3\,t_{-1}^2 t_2 & b\,t_{-1}t_2^2 \\
& & & & t_{-1}^3 & a\,t_{-1}^2 t_2
\end{pmatrix}, \tag{B.1}
$$

where we added the system size $N$ as an additional subscript, and the termination of the bottom right corner of $\mathcal{H}_{1,N}$, i.e., the right end of the chain, depends on the length $N$ of the chain as

$$
(a, b) = \begin{cases}
(2, 1) & N \bmod 3 = 0, \\
(3, 2) & N \bmod 3 = 1, \\
(3, 3) & N \bmod 3 = 2.
\end{cases} \tag{B.2}
$$

We realize that in the last case, i.e., $(a, b) = (3, 3)$, $\mathcal{H}_{1,N}$ is a Toeplitz matrix, whereas in the first two cases, the bulk is the same Toeplitz matrix with a slightly perturbed right edge. To compute the determinant, we do a Laplace expansion along the last column, and find the recurrence relation

$$
\det(\mathcal{H}_{1,N}) = (a\,t_{-1}^2 t_2)\,T_{N-1} - (b\,t_{-1}t_2^2)(t_{-1}^3)\,T_{N-2} + (t_2^3)(t_{-1}^3)(t_{-1}^3)\,T_{N-2}, \tag{B.3}
$$

where $T_N = \det(\mathcal{H}_{1,N=3q+2})$ is the determinant of the aforementioned Toeplitz matrix. In the case $N \bmod 3 = 2$, i.e., $(a, b) = (3, 3)$, the recurrence relation reads

$$
T_N = (3\,t_{-1}^2 t_2)\,T_{N-1} - (3\,t_{-1}t_2^2)(t_{-1}^3)\,T_{N-2} + (t_2^3)(t_{-1}^3)(t_{-1}^3)\,T_{N-2}, \tag{B.4}
$$

which is solved by $T_N = (t_{-1}^2 t_2)^N (N+1)(N+2)/2$. The cases $N \bmod 3 = 0, 1$ are solved by

inserting this expression and the appropriate $(a, b)$ into Eq. (B.3). We find

$$
\det(\mathcal{H}_{1,N}) = \begin{cases} (t_{-1}^2 t_2)^N & N \bmod 3 = 0, \\ (t_{-1}^2 t_2)^N (N+1) & N \bmod 3 = 1, \\ (t_{-1}^2 t_2)^N (N+1)(N+2)/2 & N \bmod 3 = 2. \end{cases} \tag{B.5}
$$

Therefore, we generally find $\det(\mathcal{H}_{1,N}) \neq 0$ thus proving that $\mathcal{H}_1$ does not have any zero-energy eigenvalues, and all zero-energy eigenvalues of $\mathcal{H}_{21}$ must originate in the interplay between the different SLs.

## C Properties of $h_j$

In order to state general properties of the $h_j$ we start by considering the generalized HN model in the site basis so that its matrix elements are given by $(\mathcal{H}_{lr}^s)_{m,n} = t_l \delta_{m,n-l} + t_{-r} \delta_{m,n+r}$ as in Eq. (A.1). $\mathcal{H}_{lr}^s$ is similar to $\mathcal{H}_{lr}$ using a permutation matrix $P$, that is, $\mathcal{H}_{21} = P \cdot \mathcal{H}_{21}^s \cdot P^{-1}$. Without loss of generality, one can choose the unique permutation matrix, which keeps the order within each SL unchanged, i.e., the $i$th site on $\text{SL}_j$ in the site basis gets mapped to the $i$th site on $\text{SL}_j$ in the transformed basis. As example, let us consider $l = 2$, $r = 1$ and $N = 8$. The model in site basis $\mathcal{H}_{21}^s$, given by Eq. (A.1), is shown in Fig C.1(a), and the model in the transformed basis $\mathcal{H}_{21}$ is shown in Fig C.1(b), where $\mathcal{H}_{21}$ and the corresponding permutation matrix $P$ are given by

$$
\mathcal{H}_{21} = \begin{pmatrix} 0 & 0 & t_{-1} & t_2 & 0 & 0 & 0 & 0 \\ 0 & 0 & 0 & t_{-1} & t_2 & 0 & 0 & 0 \\ 0 & 0 & 0 & 0 & 0 & t_{-1} & t_2 & 0 \\ 0 & 0 & 0 & 0 & 0 & 0 & t_{-1} & t_2 \\ 0 & 0 & 0 & 0 & 0 & 0 & 0 & t_{-1} \\ t_2 & 0 & 0 & 0 & 0 & 0 & 0 & 0 \\ t_{-1} & t_2 & 0 & 0 & 0 & 0 & 0 & 0 \\ 0 & t_{-1} & 0 & 0 & 0 & 0 & 0 & 0 \end{pmatrix}, \tag{C.1}
$$

and

$$
P = \begin{pmatrix} 0 & 0 & 1 & 0 & 0 & 0 & 0 & 0 \\ 0 & 0 & 0 & 0 & 0 & 1 & 0 & 0 \\ 0 & 1 & 0 & 0 & 0 & 0 & 0 & 0 \\ 0 & 0 & 0 & 0 & 1 & 0 & 0 & 0 \\ 0 & 0 & 0 & 0 & 0 & 0 & 0 & 1 \\ 1 & 0 & 0 & 0 & 0 & 0 & 0 & 0 \\ 0 & 0 & 0 & 1 & 0 & 0 & 0 & 0 \\ 0 & 0 & 0 & 0 & 0 & 0 & 1 & 0 \end{pmatrix}, \tag{C.2}
$$

respectively. Here, the $h_j$ describing the hopping between the different SLs are given by

$$
h_1 = \begin{pmatrix} t_{-1} & t_2 & 0 \\ 0 & t_{-1} & t_2 \end{pmatrix}, \qquad h_2 = \begin{pmatrix} t_{-1} & t_2 & 0 \\ 0 & t_{-1} & t_2 \\ 0 & 0 & t_{-1} \end{pmatrix}, \qquad \text{and} \qquad h_3 = \begin{pmatrix} t_2 & 0 \\ t_{-1} & t_2 \\ 0 & t_{-1} \end{pmatrix}. \tag{C.3}
$$

Overall, one can convince oneself that one can either hop from site index $i$ to $i$ and from $i+1$ to $i$, or one can hop from $i$ to $i$ and $i$ to $i+1$, cf. Fig. C.1(b). By carefully considering the individual SL sizes and hopping strengths, one finds that the matrix elements of $h_j$ are either $(h_j)_{m,n} = t_{-r} \delta_{m,n} + t_l \delta_{m,n-1}$ or $(h_j)_{m,n} = t_{-r} \delta_{m,n+1} + t_l \delta_{m,n}$.

In the main text we use that the $h_j$ describing the hopping from large SL to large SL, i.e., the $h_j$ are of size $(d_1 + 1) \times (d_1 + 1)$, are invertible. From the explicit form of $h_j$ in that case it is clear that they have full rank as $t_l, t_{-r} \neq 0$, and they are thus invertible.

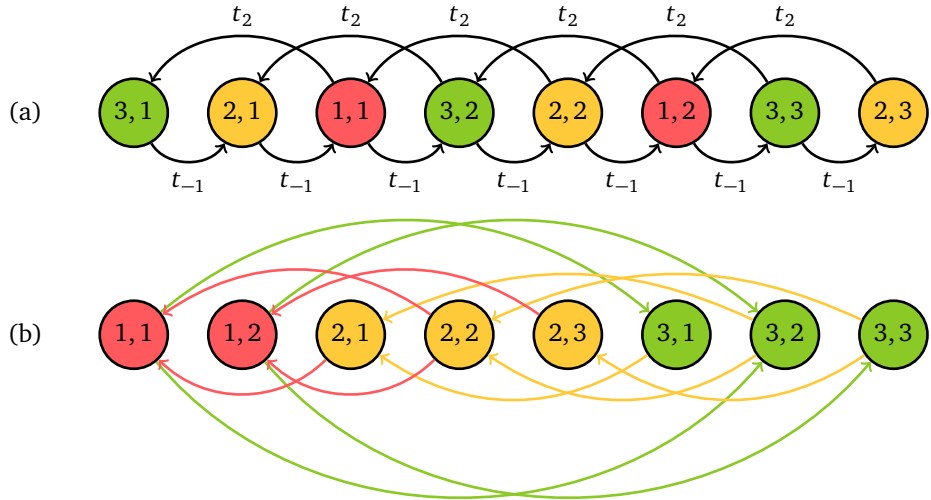

Figure C.1: Generalized HN model for $l = 2$, $r = 1$ and $N = 8$. Each node contains $j, i$, where $j$ refers to $SL_j$ and $i$ to the index within each SL. (a) Model in site basis. (b) Model in the transformed basis. Red, yellow and green arrows correspond to entries in $h_1$, $h_2$ and $h_3$, respectively. Hoppings above (below) the chain corresponds to $t_2$ ($t_{-1}$).

# D  Reduction of the order of the EPs with on-site terms

In the main text in Sec. 4.2.2, we state that on-site perturbations can be used to reduce the order of an EP. More specifically, we reasoned that for $l = 3$, $r = 1$ and $N \equiv -1 \bmod 4$, a perturbation on $SL_1$ leaves the EP3 unchanged, while a perturbation on $SL_4$ reduces the EP3 to an EP2. Furthermore, we stated that introducing on-site disorder on $SL_2$ or $SL_3$ also reduces the EP3s to EP2s. In the following, we show these statements.

For that, let us introduce a Hamiltonian which is perturbed on $SL_j$ as

$$H_{31}^{(j)} = H_{31} + H_{\text{pert}}^{(j)}, \qquad \text{where} \quad H_{\text{pert}}^{(j)} = \sum_n t_{0,n} \delta_{(n \bmod 4),(5-j \bmod 4)} c_n^\dagger c_n. \qquad \text{(D.1)}$$

For simplicity, let us set a constant on-site potential, i.e., $t_{0,n} \equiv t_0$, which is enough to see the reduction of the order of the EP while keeping the notation compact. The reason for this is that all the generalized eigenvectors of $H_{31}$ are proper eigenvectors of $H_{\text{pert}}^{(j)}$ with eigenvalue $t_0$, i.e.,

$$H_{\text{pert}}^{(j)} |\tilde{u}_0^{j'}\rangle = \delta_{j,j'} t_0 |\tilde{u}_0^j\rangle. \qquad \text{(D.2)}$$

Reminding ourselves that the Jordan chain for the unperturbed model reads $H_{31}|\tilde{u}_0^2\rangle = 0$, $H_{31}|\tilde{u}_0^3\rangle = |\tilde{u}_0^2\rangle$ and $H_{31}|\tilde{u}_0^4\rangle = |\tilde{u}_0^3\rangle$, one can immediately verify that the following equations:

- Perturbation on $SL_1$

$$H_{31}^{(1)}|\tilde{u}_0^2\rangle = 0, \qquad H_{31}^{(1)}|\tilde{u}_0^3\rangle = |\tilde{u}_0^2\rangle, \qquad H_{31}^{(1)}|\tilde{u}_0^4\rangle = |\tilde{u}_0^3\rangle.$$

- Perturbation on $SL_2$

$$H_{31}^{(2)}\left[|\tilde{u}_0^2\rangle - t_0|\tilde{u}_0^3\rangle\right] = 0, \qquad H_{31}^{(2)}\left[|\tilde{u}_0^3\rangle - t_0|\tilde{u}_0^4\rangle\right] = \left[|\tilde{u}_0^2\rangle - t_0|\tilde{u}_0^3\rangle\right], \qquad H_{31}^{(2)}|\tilde{u}_0^2\rangle = t_0|\tilde{u}_0^2\rangle.$$

- Perturbation on $SL_3$

$$H_{31}^{(3)}|\tilde{u}_0^2\rangle = 0, \qquad H_{31}^{(3)}\left[|\tilde{u}_0^3\rangle - t_0|\tilde{u}_0^4\rangle\right] = |\tilde{u}_0^2\rangle, \qquad H_{31}^{(3)}\left[|\tilde{u}_0^2\rangle + t_0|\tilde{u}_0^3\rangle\right] = t_0\left[|\tilde{u}_0^2\rangle + t_0|\tilde{u}_0^3\rangle\right].$$

- Perturbation on $SL_4$

$$H_{31}^{(4)}|\tilde{u}_0^2\rangle = 0, \quad H_{31}^{(4)}|\tilde{u}_0^3\rangle = |\tilde{u}_0^2\rangle, \quad H_{31}^{(4)}\left[|\tilde{u}_0^2\rangle + t_0|\tilde{u}_0^3\rangle + t_0^2|\tilde{u}_0^4\rangle\right] = t_0\left[|\tilde{u}_0^2\rangle + t_0|\tilde{u}_0^3\rangle + t_0^2|\tilde{u}_0^4\rangle\right].$$

In the last three cases, we see that we get a Jordan chain of length two, i.e., an EP2, while also shifting one of the zero-energy solutions to $t_0$. While this example shows the reduction of the order of the EP3 for constant on-site potentials, we report that this is true for generic on-site potentials.

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
