# Peer review of "Exceptional points of any order in a generalized Hatano-Nelson model"

_SciPost Physics_

## Round 2 · Referee Report · Anonymous (Referee 1) · 2025-1-16

Report

This paper introduces a generalized Hatano-Nelson model to systematically construct exceptional points (EPs) of arbitrary, fixed order that do not scale with system size. By extending the hopping range in the model, the authors uncover how the interplay between system size and sublattice structure determines the EP order. The eigenstates associated with these EPs exhibit the non-Hermitian skin effect (NHSE), with tunable localization properties. Additionally, the EPs are shown to be robust against perturbations in hopping strengths and certain forms of on-site disorder, underscoring their topological origin.

While the scope of the work is clear, and the presentation is technically sound, it is difficult to discern how this paper substantially advances the field beyond prior works (e.g., Refs. [22–29]). The authors propose a recipe for constructing higher-order EPs using a generalization of the Hatano-Nelson model, but the core findings appear incremental in the context of existing literature, particularly in comparison to Refs. [25–27], which also addresses higher-order EPs in non-Hermitian systems.

In my view, the results presented here do not convincingly “open a new pathway in an existing or a new research direction, with clear potential for multi-pronged follow-up work,” as required by the acceptance criteria for SciPost Physics chosen by the Authors.
Instead, the work represents an incremental extension of established concepts. While valuable, it lacks the substantial edge required for SciPost Physics.
I recommend considering this manuscript for publication in SciPost Physics Core, where it would still make a meaningful contribution to the field.

Recommendation

Accept in alternative Journal (see Report)

---

## Round 2 · Referee Report · Anonymous (Referee 2) · 2025-1-21

Report

The manuscript titled "Exceptional Points of Any Order in a Generalized Hatano-Nelson Model" submitted to \textit{SciPost Physics} by Kunst et al. is a well-written and organized article that contributes significantly to the understanding of exceptional points (EPs) of arbitrary order in a generalized Hatano-Nelson model. This model is $(l+r)$-partite with left ($t_l$) and right ($t_{-r}$) hopping between various sites on both sides. The model preserves chiral symmetry, with the EPs tied to the center of rotation.

The authors begin by discussing a specific example with particular hopping parameters ($l = 2$ and $r = 1$) before generalizing their analysis to arbitrary values of $l$ and $r$. They demonstrate the engineering of low-order EPs by shortening the Jordan chain length through the removal of the generalized eigenvector with the largest sublattice index. While they note that EPs depend on system size, they emphasize that EP properties do not scale with it. Additionally, eigenstates associated with EPs localize at one end of the system, contrasting with the localization of other eigenstates at the opposite end. Remarkably, the authors find that EPs are robust against perturbations in hopping strength and on-site random disorder in certain sublattices.

Specific comments:

  1. Clarification of Coprime Condition:

    The authors require $t_l, t_{-r} > 0$ and $l \geq r \geq 1$, with $\text{gcd}(l, r) = 1$. However, the necessity of the coprime condition is not immediately obvious. Could the authors provide a more detailed explanation or motivation for this requirement?

2 . Localization Mechanism:

The discussion regarding the localization of eigenvectors with and without EPs on opposite ends of the chain, specifically as tuned by $t_2$ and $t_{-1}$, is not entirely clear. Furthermore, the condition for localization, $|E_B| > |E_s^{1/3}|$ or $|E_B| < |E_s^{1/3}|$, requires elaboration to enhance the reader's understanding.

  1. Robustness to Disorder:

The authors claim that random disorder in specific sublattices does not affect EP behavior. Is this conclusion valid for all types of disorder (e.g., quasiperiodic) and for disorder affecting any sublattice? Additional clarification or discussion on this aspect would strengthen the manuscript.

  1. Language and Style:

    The manuscript contains repetitive phrases and grammatical errors, particularly in the abstract and other sections. A thorough revision to eliminate redundancies and improve clarity is recommended for better readability and impact.

Attachment

Recommendation

Ask for minor revision

---

## Round 2 · Referee Report · Anonymous (Referee 3) · 2025-2-19

Report

This manuscript studies the spectral properties and the associated exceptional points (EPs) of a generalised Hatano-Nelson (HN) model of non-Hermitian quantum mechanics. The generalisation involves asymmetry in the hopping range between left and right directions (as opposed to asymmetry in the hopping strength as in the standard HN model).
The Authors discuss the presence of exceptional points in the spectrum of the model, first in a specific case (See Eq. 2) and then in more general terms.
The main result of this work is that the generalised HN model features EPs of order m<N (with N the system size). Finally, they characterize the EPs via their Skin effect and their robustness to perturbations such as disorder.

The manuscript contains interesting results which are worth publishing. I do not see however the acceptance criteria of Scipost Physics to be met by this manuscript. The specific result of this work - finding EPs of order m<N - does not qualify in my opinion for a breakthrough or a groundbreaking result by itself.
It could be of interest for a certain community working on non-Hermitian spectral properties, but not more broadly.

Furthermore, the manuscript is written in a very technical way which makes it hard to appreciate the implications of these results. There is a lot of emphasis on the derivation of the main mathematical statements while not much on the interpretation and consequences. The figures with the major results on the properties of the EPs (Fig 6-7) are not even discussed in detail in the main text (only left to the caption).

To conclude I think the Authors should largely improve the presentation, giving more space to discuss the results presented in support of their mathematical reasoning. Once this is done I think the paper could qualify for Scipost Core according to its acceptance criteria ("one or more new research results significantly advancing current knowledge and understanding of the field.")

Recommendation

Reject

---

## Editorial Decision

awaiting_resubmission